# Adaptation and Virulence of Enterovirus-A71

**DOI:** 10.3390/v13081661

**Published:** 2021-08-21

**Authors:** Kyousuke Kobayashi, Satoshi Koike

**Affiliations:** Neurovirology Project, Tokyo Metropolitan Institute of Medical Science, Tokyo 156-8506, Japan; kobayashi-ks@igakuken.or.jp

**Keywords:** enterovirus-A71, virulence determinant, adaptation, mutation, infection animal model

## Abstract

Outbreaks of hand, foot, and mouth disease caused by enterovirus-A71 (EV-A71) can result in many deaths, due to central nervous system complications. Outbreaks with many fatalities have occurred sporadically in the Asia-Pacific region and have become a serious public health concern. It is hypothesized that virulent mutations in the EV-A71 genome cause these occasional outbreaks. Analysis of EV-A71 neurovirulence determinants is important, but there are no virulence determinants that are widely accepted among researchers. This is because most studies have been done in artificially infected mouse models and because EV-A71 mutates very quickly to adapt to the artificial host environment. Although EV-A71 uses multiple receptors for infection, it is clear that adaptation-related mutations alter the binding specificity of the receptors and allow the virus to adopt the best entry route for each environment. Such mutations have confused interpretations of virulence in animal models. This article will discuss how environment-adapted mutations in EV-A71 occur, how they affect virulence, and how such mutations can be avoided. We also discuss future perspectives for EV-A71 virulence research.

## 1. Epidemiology of Enterovirus-A71 (EV-A71)

Enterovirus A (EV-A) belonging to the genus *Enterovirus* (family, *Picornaviridae*) has various serotypes. Serotypes, such as Coxsackievirus A6 (CVA6), CVA10, CVA16, and EV-A71, cause hand, foot, and mouth disease (HFMD). Usually, HFMD is an infectious disease that mainly affects infants and young children, with the primary symptom being a bullous rash on the oral mucosa, hands, and feet. Unlike other serotypes, EV-A71 can sometimes cause central nervous system (CNS) complications, such as meningitis, cerebellar ataxia, acute flaccid paralysis, encephalitis, and pulmonary edema. Outbreaks of HFMD, in which many patients develop CNS complications caused by EV-A71, have occurred sporadically in the Asia-Pacific region, including Malaysia, Taiwan, Mainland China, Vietnam, and Cambodia since the late 1990s [1,2,3,4,5,6,7,8,9,10,11]. It is an infectious disease with serious public health implications because a large-scale outbreak can result in many deaths. 

## 2. Viral Replication

EV-A71 has a single-stranded plus-stranded RNA genome [12] of about 7400 bases encapsulated by a non-enveloped capsid. The genome encodes a single polyprotein of about 2200 amino acid residues, cleaved into three precursor proteins (P1, P2, and P3) after translation. These are cleaved further to yield 11 different viral proteins. Four capsid proteins arise from P1, and seven nonstructural viral proteins arise from P2 and P3. The polyprotein coding region is flanked by 5′- and 3′-untranslated regions (UTRs). Functional RNA structures on the UTRs, such as internal ribosome entry sites (IRES) and cloverleaf, cooperate with viral proteins and host proteins to translate viral proteins and replicate the RNA genome. Therefore, amino acid mutations in viral proteins may alter their activity, or nucleotide mutations in the UTRs may alter translation and replication efficiency, thereby altering virulence.

Virus receptors on the cell surface mediate the early infection steps: Virion attachment to the host cell surface; internalization into the cell; and uncoating and releasing genomic RNA from the capsid into the cytoplasm. Scavenger receptor class B member 2 (SCARB2), a well-studied receptor for EV-A71, can mediate these steps [13,14]. Co-crystallographic analysis with EV-A71 particles and SCARB2 revealed that several amino acids in the VP2 EF-loop and VP1 GH-loop on the virion are involved in the binding to SCARB2 [15]. However, since SCARB2 is naturally a lysosomal protein, its expression at the cell surface is low. Therefore, under certain conditions, receptors other than SCARB2, called attachment receptors, may assist the virus, and enable efficient infection. Attachment receptors include P-selectin glycoprotein ligand 1 (PSGL1) [16], heparan sulfate (HS) [17], sialic acid [18], annexin II [19], nucleolin [20], vimentin [21], and fibronectin [22]. Among these, HS and PSGL1 are well described. HS is a highly sulfated glycosaminoglycan biosynthesized in cells as a proteoglycan bound to a protein backbone. HS proteoglycans, which play a physiological role by binding to various ligand molecules, are expressed on the surface or secreted extracellularly [23]. PSGL1 is a type I transmembrane protein expressed on the surface of neutrophils, monocytes, and most lymphocytes. Tyrosine sulfation near the NH_2_-terminus of PSGL1 is necessary for EV-A71-binding [24]. EV-A71 particles bind to human PSGL1, but not mouse PSGL1 [16]. Among these attachment receptors, HS plays an important role in the adaptation of EV-A71 in cell culture.

After the viral genome is released into the cytoplasm, translation of viral proteins occurs in an IRES-dependent manner, and replication of the viral genome occurs via RNA-dependent RNA polymerase (RdRp). The RdRp of RNA viruses, including EV-A71, shows low fidelity; therefore, replication errors occur with high frequency, resulting in progeny harboring diverse nucleotide substitutions. This diversity of the viral genome allows more adapted variants to replicate selectively and become the dominant species; this process drives the adaptation of viruses to their environment.

## 3. Studies on Pathogenicity

As with many viral infections, some patients become severely ill, whereas others do not; likewise, some epidemics are associated with a high frequency of severe cases, and others are not. Certain factors are thought to be associated with disease severity. Candidates include the environment of the epidemic area (climate, topography, and culture), host factors (susceptibility and immunity), and viral factors (polymorphisms in nucleotides and amino acids). It is not currently clear which of these factors is more critical. However, even though EV-A71 has spread worldwide, outbreaks are sporadic. Although the environment and host factors in an epidemic area do not change over a short period, the genome of RNA viruses changes quickly, so we can assume that mutations in the viral genome are associated with outbreaks. Various attempts have been made to identify the virulence determinants on the EV-A71 genome. However, the virulence determinants currently proposed are not widely accepted by EV-A71 researchers for the following reasons: (1) There are few analyses using virulence evaluation systems that reflect human pathogenesis; (2) EV-A71 can mutate easily in cultured cells and animals and adapt to the given environment. Therefore, it is difficult to distinguish virulence determinants from adaptation determinants. These two factors are discussed in detail below.

### 3.1. Evaluation of EV-A71 Virulence

Evaluation of EV-A71 virulence includes analysis of clinical data from human patients, cynomolgus macaques, suckling mice, immunodeficient mice, and hSCARB2-transgenic (tg) mice. In addition, mouse-adapted strains obtained by repeated passages in mice have been used for virulence study combined with suckling mice or immunodeficient mice. The patient severity approach is performed with the expectation that there will be an association between disease severity in a human patient and the sequence of the viral genome isolated from that patient [25,26,27]. However, most people infected with EV-A71 have asymptomatic or mild disease, and severe cases are extremely rare; therefore, many patients do not become severely ill, even though they are infected with a highly virulent strain. Even if they develop a disease, its severity is influenced by factors, such as the host’s genetic background, nutritional status, living environment, and the amount of virus to which they are exposed. Identifying virulence determinants in association studies requires the accumulation of a large amount of data. Alternatively, animal model experiments should be conducted by infecting a relatively large number of animals with large amounts of virus (sufficient to cause apparent disease). In this way, a closer association between disease severity in infected animals and virulence determinants in the viral genome will be demonstrated. EV-A71 infects humans as its natural host, although non-human primates, such as cynomolgus monkeys, can be infected experimentally. When cynomolgus monkeys are intravenously inoculated with EV-A71, they show symptoms, such as paralysis, due to infection of the CNS; however, they do not show skin lesions like HFMD [28]. One caveat to this approach is that although this evaluation system facilitates assessment of the neurovirulence of EV-A71, the availability of experimental facilities for infecting large animals is limited, and the cost per animal and attendant ethical issues make quantitative studies difficult.

Mice are useful model animals because there are fewer problems in terms of equipment, cost, and ethics; also, they are easier to modify genetically. However, there is another problem related to the use of non-primate animal models. EV-A71 cannot use mouse SCARB2 as a receptor [29], and adult mice are not susceptible to infection by EV-A71. However, suckling mice are susceptible within one week of birth; these mice can be used as an animal model for EV-A71 infection [30,31]. Mouse models are important experimental systems that can yield a variety of experimental results. Nevertheless, it is difficult to interpret the data, particularly whether various viral genomic mutations are simply an adaptation to the mouse environment or mutations that increase virulence in humans. Unlike a human disease, EV-A71 intraperitoneally or subcutaneously inoculated to suckling mice mainly targets skeletal muscle, which does not reflect human pathology. In addition, no receptors have been identified for this infection. The immunologically modified mouse strain, AG129, an interferon α/β receptor, and interferon γ receptor double knockout mouse, is also used for virulence analysis of EV-A71 [32]. AG129 mice are more sensitive to EV-A71 than wild-type mice and are susceptible to infection until two weeks old. The virus replicates mainly in skeletal muscle and then reaches the CNS, resulting in neurological symptoms, such as flaccid paralysis. Studies are conducted in combination with the AG129 mouse model and EV-A71 mouse-adapted strains established by repeated passages in rodent cells or animals [30,33,34]. This approach can extend the window of susceptibility by more than six weeks [34]. 

A tg mouse expressing human SCARB2 can overcome the problems of the above models. EV-A71 can infect hSCARB2-tg mice of any age by intravenous, intraperitoneal, and intracerebral inoculation [35]. CNS neurons are the main target of EV-A71, causing CNS symptoms, such as paralysis, but not causing skin lesions, such as HFMD. Therefore, the hSCARB2-tg mouse is an excellent animal model for CNS disease caused by EV-A71 infection. There are reports of various hSCARB2-tg mice with different promoters and knock-in mice [36,37,38,39]. However, the exact differences in susceptibility and infection pathology among these mouse strains are not clearly described [40]. Among these hSCARB2-tg mouse models, hSCARB2-tg mice by Fujii et al. [35] are susceptible to EV-A71 at any age, and the muscle-tropic infection observed in the neonatal period is not observed in adults and is an ideal model for future virulence analysis of EV-A71, as it more closely mimics human pathogenesis and allows for quantitative analysis.

### 3.2. HS-Binding Mutations and Tissue Culture Adaptation

One of the interesting findings related to virulence changes and receptor adaptation is mutations in an HS-binding phenotype. Tan et al. [17,41] found that some mutant strains of EV-A71 can use HS as an attachment receptor. The mutant viruses can bind strongly to HS. The mutations required for the HS-binding phenotype are VP1-E145G or E145Q mutation, along with mutations at VP1-97, VP1-98, and VP1-244 [41,42,43]. These mutations increase the infection efficiency of cultured cells significantly by enhancing virus attachment to the cell surface (Figure 1). Our previous study shows that the virus infection efficiency (copy number/50% tissue culture infectious dose; TCID_50_) in RD-A cells was 47,619 copies/TCID_50_ for VP1-145E, whereas it was 200 and 100 copies/TCID_50_ for VP1-145G and VP1-145Q, respectively [44]. This result means that VP1-145G and VP1-145Q viruses infect RD-A cells 200 times more efficiently than the VP1-145E virus. VP1-145 is located on the particle surface near the 5-fold symmetry axis. This area is positively charged when the amino acid is G or Q but less positively charged when the amino acid is E (this is because the negative charge of the glutamic acid residue cancels the surrounding positive charge) [45]. Since HS is negatively charged by sulfation, it is thought to bind to G and Q particles via electrostatic interactions. Therefore, it is likely that VP1-145G and VP1-145Q viruses infect cultured cells more efficiently than the VP1-145E virus, due to the acquisition of the HS-mediated infection pathway. Other HS-binding mutations around the 5-fold symmetry axis are known [41,42,43], and may show the same phenotype as VP1-145G and Q. It should be noted that such mutations appear during replication in cultured cells because of the low fidelity of RdRp and that HS-binding mutants are rapidly selected and become predominant in the population under conventional cell culture conditions [44]. Adaptation to cultured cells is thought to occur quickly when an HS-adapted mutant happens to appear as a founder, but if the founder does not appear within the culture period, adaptation does not occur. The emergence of adapted mutants is determined stochastically. Nishimura et al. [45] found that PSGL1 also serves as an attachment receptor in Jurkat cells. Similar mutations of the viral capsid are required for PSGL1-binding because mutant viruses bind to sulfated PSGL1 via electrostatic interactions. 

To better understand the relationship between HS-binding mutations and EV-A71 virulence, we have summarized past papers (see Table 1). The increased infection efficiency of HS-binding strains may increase virulence in vivo. With this in mind, Cordey et al. [46] and Tseligka et al. [42] reported the isolation of the VP1-97L mutant from cerebrospinal fluid, stool, and plasma of an immunocompromised patient. The mutation conferred HS-binding ability and a replicative advantage in human neuroblastoma cells. However, more studies are needed to examine the virulence of the isolated virus by some in vivo experimental methods. Analysis of the relationship between disease severity and the viral genome sequence showed that infection by VP1-145G/Q/R was significantly high in severe cases [25,26,27]. However, the weakness of these reports is that the methods of virus isolation and sequencing are not described. Virus strains isolated from human specimens and grown in conventionally cultured cells undergo a rapid change into HS-binding strains. Further careful analysis is expected to reveal whether HS-binding mutations are associated with virulence in humans.

By contrast, many experimental animal studies refute the idea that HS-binding strains are virulent. For example, Chua et al. [30] and Wang et al. [31] used mice to passage EV-A71 strains that are not virulent in mice and obtained strains that caused disease. Chua et al. found that a G to E mutation at VP1-145 (HS-binding to HS-nonbinding mutation) is essential for increased virulence in neonatal mice. Similarly, Wang et al. passaged the 4643 strain in mice and obtained mouse-adapted strain MP4 [31]. By constructing a chimeric virus between 4643 and MP4, Huang et al. [33] found that two mutations co-operatively increased virulence in mice. One of the mutations is VP1-G145E. In addition to VP1-145, other HS-binding mutations associated with EV-A71 attenuation, such as VP1-E98K [43] and VP1-E244K [34,52], are known. All these mutants showed attenuated phenotypes in suckling mice. 

Kataoka et al. [49] compared the virulence of PSGL1-binding and -nonbinding strains in cynomolgus monkeys. The PSGL1-binding strains used in this study were also HS-binding strains. The results showed that the PSGL1/HS-binding strains are less virulent than the nonbinding strains; they soon disappeared from the infected monkeys; and were replaced by a nonbinding strain. Similarly, Fujii et al. [51] showed that the HS-nonbinding strain is more virulent than the binding strain. Furthermore, Kobayashi et al. [50] showed a similar result using hSCARB2-tg mice. Thus, HS-nonbinding strains are more virulent than HS-binding strains. Viruses converted to HS-nonbinding strains were recovered from the CNS of paralyzed mice inoculated with a high dose of an HS-binding strain. The authors proposed the following hypothesis: In hSCARB2-tg mice and cynomolgus monkeys (and probably in other animals), cells expressing HS at very high levels (e.g., vascular endothelial cells, sinusoidal endothelial cells, and glomeruli) do not express SCARB2, whereas cells expressing SCARB2 (e.g., neurons, hepatocytes, and tubular epithelial cells) do not express HS [50,51]. HS is also present in the extracellular matrix. HS-binding viruses cannot infect their primary target cells in this situation because extracellular matrices and cells that do not express SCARB2 adsorb the HS-binding viruses [50]. Therefore, unlike cultured cells in vitro, HS does not support virus replication in vivo. Instead, it inhibits the dissemination of the virus by adsorbing the viruses. Tee et al. [52] also observed a similar adsorption phenomenon in suckling mice infected with EV-A71 possessing HS-binding amino acids at VP1-145 and VP1-244. In addition, an adsorption phenomenon was reported in an EV-A71 strain carrying the mutation VP1-E98K [43]. Therefore, any amino acid mutation in the HS-binding type is likely to reduce the virulence of EV-A71 in animal models. 

### 3.3. Mouse (Rodent) Adaptation

Except for hSCARB2-tg mice, infection with most EV-A71 wild-type strains is limited to suckling mice aged up to 1 week. Some research groups have tried to increase virulence in mice to elucidate the mechanisms underlying increased virulence in mice and humans. We have collected studies reporting mouse adaptation and listed them in Table 2. Chua et al. [30] and Wang et al. [31] identified mutations at VP2-149 that increased virulence in suckling mice. They found that the VP2-149 mutation K to I or M, and the VP1-145 mutation G to E, co-operatively contribute to disease. There are also reports of isolating the VP2-149I virus, a highly virulent strain in mice, from human samples without mouse adaptation [55]. This result suggests that the VP2-149I mutant is circulating in humans at a low frequency. It is unclear from these in vivo results whether these mutants merely adapted in mice or whether they show increased virulence in mice and humans. 

EV-A71 naturally infects and replicates in human cells, but the infection efficiency in mouse cells is extremely low because EV-A71 cannot use mouse SCARB2 as a receptor [29]. Therefore, attempts have been made to grow the EV-A71 strains in rodent cells. Chua et al. [30], Victorio et al. [57], and Zaini et al. [48] showed independently that adapted mutants could be obtained by serial passage in hamster or mouse cell lines. The mutants contained the VP2-K149I or VP2-K149M mutation. These mutations increase infectivity and proliferation in mouse and hamster cell lines [48,57]. Therefore, these mutations are considered to be the result of rodent adaptation. 

In addition, Miyamura et al. [56] performed an interesting adaptation experiment. Mouse L929 cells overexpressing PSGL1 (L-PSGL1) are not fully susceptible to EV-A71 (PSGL1-binding strain), and the viruses replicate very slowly. They obtained mutant strains adapted to L-PSGL1 cells by passaging five different PSGL1-binding strains for one generation in L-PSGL1 cells. As a result, they obtained adapted strains that replicate with normal kinetics, four strains with the VP2-149 mutation, and one strain without the VP2-149 mutation. Since PSGL1 is an attachment receptor that cannot initiate the uncoating reaction, the mutation at VP2-149 is thought to enhance the uncoating reaction in rodent cells. Co-crystallographic analysis of EV-A71 and human SCARB2 revealed hydrophobic interactions between VP2-149 and four residues on SCARB2 alpha helix 5: 158A, 159M, 162A, and 163Y (Figure 2) [15]. At that region, the sequence of human and mouse SCARB2 is most divergent [29]. The result suggested that the amino acid mutation at VP2-149 is associated with altered receptor usage and the acquisition of uncoating activity in rodent cells. 

Besides the mutation at VP2-149 described above, Victorio et al. [58] reported that VP1-K98E, E145A, and L169F use mouse SCARB2. Interestingly, a single amino acid mutation in VP1-L169F gives it infectivity in a mouse cell line. Although these amino acids are not directly binding sites for SCARB2 [15], these changes may indirectly affect the binding to mouse SCARB2. However, little analysis of this amino acid has been done by other groups, and further analysis is required. Miyamura et al. [56] also obtained a mutant adapted to L-PSGL1 cells without the VP1-K149I mutation. This mutant contained VP2-K69R, VP2-V135I, VP2-T176P, VP1-X145Q (where X is a mixture of two or more amino acids), and VP1-I249V. Rodent adaptation may occur in several different mechanisms.

It is unclear whether this mutation allows mouse or hamster SCARB2 to be used as an uncoating receptor or allows other receptors to be used. However, clarifying the molecular mechanism by which this amino acid mutation alters the properties of the virus particle and why it can infect mice makes a significant contribution to clarifying some of the factors necessary for the virulence of EV-A71. It is important to remember that this mutation is likely an adaptation determinant, not a virulence determinant. However, further studies should examine whether this mutation explains the severity of the disease in humans.

Figure 3 combines the results of VP1 and VP2 mutations. The interpretation of some original articles is confusing and misleading. However, it is possible to interpret the results from two axes. One axis comprises mutations controlling the HS-binding phenotype. HS-binding and HS-nonbinding phenotypes are associated with in vitro cell culture adaptation and in vivo adaptation, respectively. HS-binding strains grow more efficiently in cultured cells, but HS-nonbinding strains replicate more efficiently in an animal body and cause severe disease. The other axis is the host range-affecting mutation at VP2-149. VP2-149I/M mutants can infect rodents, probably using the mouse SCARB2 protein as an uncoating receptor (though it has not been demonstrated). It has not been proven whether the VP2-149 mutation is virulent in humans.

### 3.4. Possible Neurovirulence Determinants Not Influenced by Adaptation Mutations

So far, we have discussed mutations involved in the adaptation of cultured cells and mice; however, there are also reports of virulence determinants that are not related to these mutations. They can be classified into mutations associated with the fidelity of viral RdRp, mutations related to temperature sensitivity (*ts*), and mutations that do not belong to any of these categories. The former two were produced artificially. These attempts may be important for developing live attenuated vaccines. The third appeared spontaneously in a human patient. Here, we discuss how these mutations affect EV-A71 virulence.

As mentioned already, the fidelity of RNA virus RdRp is low. However, poliovirus (PV) research revealed that 3D-G64S is an amino acid mutation that increases fidelity. This mutation reduces PV virulence [59,60,61]. Meng and Kwang showed an association between EV-A71 virulence and RdRp fidelity [62]. They cultured EV-A71 in the presence of ribavirin, a mutagenic reagent, and obtained ribavirin-resistant EV-A71 mutants. The mutant acquired the 3D-L123F mutation, which increases the replication fidelity of the RdRp. They also produced various 3D-64 mutants based on the PV study and identified 3D-G64R as a viable mutant; they showed that this mutation contributed to the increase in replication fidelity. In addition, the G64R and L123F double mutant showed even greater replication fidelity. The L123F mutation showed a reduced growth rate in cultured cells, but G64R was comparable to the wild type. When these single and double mutants were inoculated into AG129 mice and tested for virulence, all were less virulent than the wild type, with the double mutant being the weakest. The results show that increased replication fidelity reduces the virulence of EV-A71, which is consistent with the report for PV. 

*Ts* is an essential phenotype of live attenuated PV vaccine strains (Sabin 1, 2, and 3), which prevents the virus from replicating at high temperatures (38–40°C). Mutations in the PV genome associated with *ts* exist across multiple locations [63]. Several reports have examined *ts* mutations in EV-A71. Arita et al. [64] compared the sequence of *ts* with that of temperature-resistant mutants of the EV-A71 reference strain, BrCr, and found a 9-nucleotide mismatch. When analyzing these mutant viruses, they found that VP1-Y116H was associated with *ts*. In addition, using known nucleotide and amino acid mutations that define the difference in virulence between the virulent strains of PV1 Mahoney and the attenuated strain Sabin1, they found that mutations in 5’-UTR-485, 3D-73, 3D-363, and 3’-UTR-7409 make a combined contribution to *ts*. In addition, these *ts*-mutant strains show reduced virulence in cynomolgus monkeys. In addition, Kung et al. [65] showed that 3D-251 plays an important role in the *ts* of EV-A71. Both the 3D-251I and T mutants showed similar growth at 35°C, but at 39°C, the growth rate of 3D-251T was reduced markedly, with a slight decrease in virulence in suckling mice.

The mutations mentioned so far allow us to estimate the underlying mechanism for changes in virus properties. However, the following mutations are also interesting, although the putative mechanism(s) is unknown. Huang et al. [66] performed a haplotype analysis of viral genomes in various organs of a fatal case of EV-A71 infection. The genetic diversity of EV-A71 in the respiratory and gastrointestinal tract lumens was extremely high, but specific haplotypes were selected to replicate in the CNS. Only VP1-D31G was a positively selected amino acid mutation in CNS. In vitro analysis showed that VP1-31G particles were less stable than VP1-31D particles and showed high proliferation in neuron-derived cell lines. This result suggests that viruses that spread from human to human and viruses that reach the CNS after infecting humans have different genetic characteristics. However, the difference in neurovirulence has not been verified in animal experiments. More detailed analysis of this amino acid mutation and more clinical data are needed to support it.

## 4. Identification of True Virulence Determinants

As we described above, the systems used to evaluate the virulence of EV-A71 have many problems. When a virus from a human clinical specimen is transferred to RD-A and Vero cells, both of which are used widely for EV-A71 culture, mutations of the HS-binding phenotype occur with a high frequency and are strongly selected [44]. To identify the true virulence determinants of EV-A71, we need to optimize virus isolation and propagation methods, as well as virulence evaluation methods. Therefore, we established a cell line, RD-∆EXT1+hSCARB2, lacking an enzyme involved in HS biosynthesis (exostosin glycosyltransferase 1) and overexpressing human SCARB2. In this cell line, the selection of HS-binding strains can be minimized (Figure 1). Since even contamination of a virus sample with a small amount of HS-binding mutant can affect the viral titer significantly, the virus stock used in the experiment should be verified for the presence of HS-binding mutations using a next-generation sequencer. To the best of our knowledge, the best model for evaluating viral virulence is the hSCARB2-tg mouse, which shows very similar neurological symptoms to humans. This method allows the evaluation of clinically isolated strains without any bias for host range adaptation and enables the use of sufficient mice for quantitative analysis.

We would like to introduce the virulence evaluation experiments we have conducted recently, following this analysis flow [67]. We conducted an epidemiological study of the HFMD epidemic in Vietnam in 2015–2016. The results showed that EV-A71 was prevalent in subgenogroups B5 and C4 simultaneously and that the severity (hospitalization rate) was higher in patients infected with C4. To confirm this experimentally, we isolated the virus by infecting EV-A71-positive patient specimens (pharyngeal or rectal swabs) with RD-∆EXT1+hSCARB2. After conducting next-generation sequencing to confirm that none of the strains used in the infection experiments contained all known HS-binding and mouse-adapted mutations, the viral titer (TCID_50_) was quantitated using RD-A cells. The 6–7-week-old hSCARB2-tg mice were inoculated intraperitoneally with the isolated strains (500,000 TCID_50_/mouse) and were evaluated for CNS symptoms (paralysis of limbs) and mortality. The results showed that C4 was more virulent, as was the patient hospitalization rate. By following this flow, it is possible to perform virulence analysis using a system that mimics human pathology while maintaining the viral genome sequence in the patient. We also propose defining the viral dose for inoculation by RNA copy number or the number of infectious particles. In this case, one can avoid the problem that the relative titer of inoculation virus varies depending on the cell lines used for titration. Further studies based on carefully selected examination systems should identify virulence determinants of EV-A71 in the future. 

## Figures and Tables

**Figure 1 viruses-13-01661-f001:**
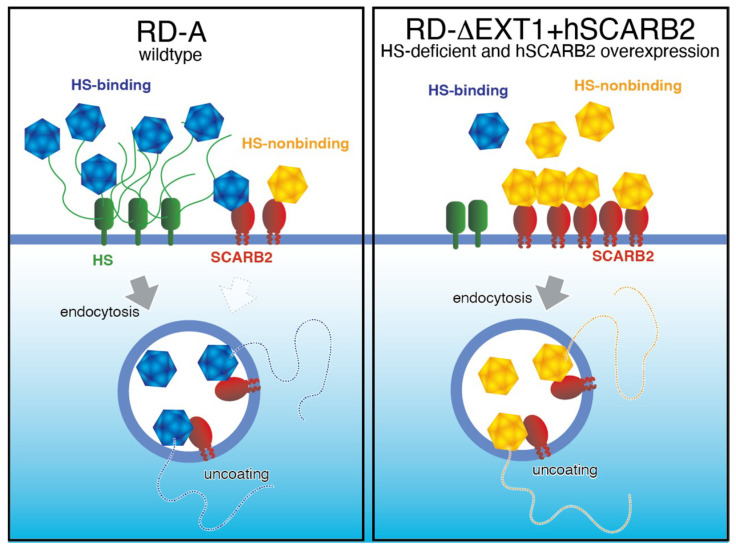
Selective infection of cultured cells by HS-binding mutants. (Left panel) Normal cell lines, such as RD-A, show low cell surface expression of SCARB2 and high expression of HS. Therefore, the infection efficiency of wild-type HS-nonbinding mutants is low. By contrast, a small number of HS-binding mutants emerge during replication and efficiently infect cells through HS; thus, HS-binding mutants become dominant. (Right panel) In RD-∆EXT1+hSCARB2 cells, selective infection by HS-binding mutants is less likely to occur because the expression of SCARB2 and HS, responsible for such infection bias, is optimized.

**Figure 2 viruses-13-01661-f002:**
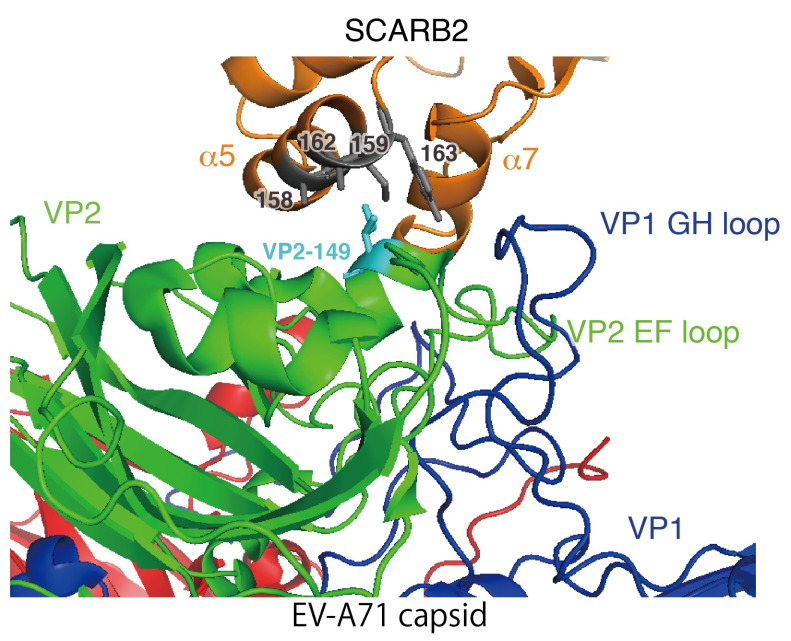
The binding site of VP2-149 and hSCARB2. The 3D structure of the EV-A71 capsid protomer (VP1, VP2, and VP3 in blue, green, and red, respectively) and the ectodomain of hSCARB2 (orange) are shown in cartoon representation. VP2-149 is shown as a cyan stick, and hSCARB2 residues 158A, 159M, 162A, and 163Y are shown as gray sticks. This figure was produced using Protein Data Base 6I2K.

**Figure 3 viruses-13-01661-f003:**
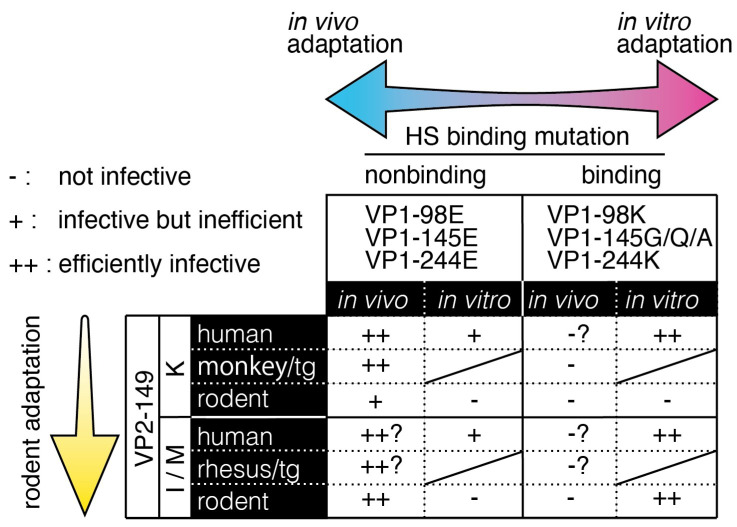
The HS-binding mutation controls in vitro (cultured cells) and in vivo adaptations, whereas the VP2-149 mutation controls adaptation to rodents. Human/in vivo indicates infection of human patients, human/in vitro indicates infection of human-derived cell lines, monkey/tg/in vivo indicates infection of cynomolgus monkeys and hSCARB2-tg mice, rodent/in vivo indicates infection of suckling mice, and rodent/in vitro indicates infection of rodent cell lines.

**Table 1 viruses-13-01661-t001:** Studies on HS-binding mutations.

Amino Acid Position	Amino Acid Residue		Description	References
	Virulent	Avirulent		
VP1-145	G/Q/R	E	VP1-145G/Q/R was more frequently detected in severe human cases than VP1-145E.	[25]
VP1-145	Q	E	VP1-145Q was found in two of nine isolates from severe human cases, but all mild cases were VP1-145E.	[26]
VP1-145	Non-E	E	VP1-145 non-E (the actual amino acid residues are not shown in the paper) was more frequently detected in HFMD severe cases than VP1-145E.	[27]
VP1-97	R	L	VP1-97R was isolated from cerebrospinal fluid, stool, and plasma of an immunocompromised patient. VP1-97R conferred a replicative advantage in a human neuroblastoma cell line, SH-SY5Y. VP1-97R conferred HS-binding ability.	[42,46]
VP1-98	E	K	VP1-E98K was acquired by 30 passages of a mouse-adapted strain in a mouse cell line, L929. VP1-E98K conferred a replicative advantage in mouse cell lines L929 and Neuro2A. VP1-E98K enhanced binding to HS. VP1-E98K attenuated the virulence in 2-week-old BALB/c mice.	[43]
VP1-145	E	G	VP1-G145E was acquired by six passages of Chinese Hamster Ovary (CHO) cells-adapted strain in 1-day-old BALB/c mice. VP1-G145E enhanced virulence in 1-day-old BALB/c mice.	[30]
VP1-145	E	G	VP1-G145E was acquired by three passages in 3-week-old NOD/SCID mice. VP1-G145E enhanced virulence in 3-week-old NOD/SCID mice.	[47]
VP1-145	E	Q	VP1-Q145E was introduced in a C4 strain and enhanced virulence in 5-day-old BALB/c mice.	[48]
VP1-145	E	Q	VP1-Q145E was acquired after four passages in 1-day-old ICR mice. VP1-Q145E and VP2-K149M co-operatively enhanced virulence in 1-day-old ICR mice.	[31,33]
VP1-145	E	G	VP1-145E is virulent, but VP1-145G is avirulent in cynomolgus monkeys.	[49]
VP1-145	E	G	VP1-145G proliferates well in cell lines, such as RD and L-SCARB2, whereas VP1-145E does not. VP1-145G binds well to HS, whereas VP1-145E does not. VP1-145E is virulent, but VP1-145G is avirulent in 6–7-week-old hSCARB2-tg mice. VP1-145G is unable to disseminate in the mouse body and reach the CNS.	[50]
VP1-145	E	G	VP1-145E is virulent, but VP1-145G is avirulent in cynomolgus monkeys. VP1-145G is easily neutralized by antibodies, but VP1-145E is not.	[51]
VP1-145 VP1-244	E E	Q K	VP1-145Q binds well to HS, whereas VP1-145E does not. VP1-145E is virulent, but VP1-145G is avirulent in 1-day-old ICR mice. The VP1-K244E mutation was found in the brains of mice infected with VP1-145Q and was developed (VP1-145Q/244E). VP1-145Q/244E showed low HS-binding and high mouse virulence.	[52]
VP1-145	E	G/Q	VP1-E145G/Q occurs during growth in cultured cells. VP1-E145G/Q does not occur when growing in HS-deficient hSCARB2 overexpressing cells (RD-∆EXT1+hSCARB2).	[44]
VP1-244	E	K	VP1-K244E was detected in a strain that was passaged three times in AG129 mice. VP1-244E is virulent, but VP1-244K is avirulent in 6-week-old AG129 mice.	[34,53]
VP1-244	E	K	VP1-K244E was detected in a strain that was passaged five times in 1-day-old BALB/c mice. VP1-244E is virulent, but VP1-244K is avirulent in 5-day-old BALB/c mice.	[54]

**Table 2 viruses-13-01661-t002:** Studies on mouse adaptation.

Mouse Adaptation Mutation	Adapted Strain	Adaptation Procedure	Replication in Cell Lines	Virulence in an Animal Model	References
VP2-K149M	MP4	Four passages in 1-day-old ICR mice	MP4 is highly proliferative in several human cell lines	VP2-K149M and VP1-Q145E are together responsible for mouse virulence	[31,33]
VP2-K149I	CHO-26M MP-26M	Six passages in a hamster cell line (CHO), then f our passages in suckling mice	NT	VP2-K149I did not contribute much, and VP1-G145E was the most critical mutation	[30]
VP2-K149I VP2-K149M	1095-LPS1 SK-EV006-LPS1 C7/Osaka-LPS1 75-Yamagata-LPS1	One passage in a human PSGL1 overexpressing mouse cell line (L-PSGL1)	Adaptation increased proliferation in L-PSGL1	NT	[56]
VP2-K149I VP2-K149M	CHO-B5 CHO-C2	Four to eight passages in a hamster cell line (CHO)	CHO-B5, CHO-C4, and the parental strains containing VP2-K149I or VP2-K149M showed enhanced proliferation in CHO cells. All of these viruses are VP1-145Q (HS-binding)	NT	[48]
VP2-K149I	EV71:TLLm EV71:TLLmv	60 and 100 passages in a mouse cell line (NIH/3T3)	TLLm and TLLmv show increased efficiency for infecting various rodent cell lines, but the VP2-K149I point mutant does not show the same increase in infection in such cells. The reason for this may be that the strains used are HS-nonbinding	No increase in virulence was observed with VP2-K149I	[57,58]
VP2-K149I	GZ-CII	VP2-149I have been isolated from a human patient	NT	GZ-CII and artificially mutated VP2-K149I viruses are highly virulent in mice	[55]

NT: not tested.

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
