# Peer review of "Adaptation and Virulence of Enterovirus-A71"

_viruses, 2021, doi:10.3390/v13081661_

Round 1

Reviewer 1 Report

The review by K. Kobayashi and S Koike aims to assess the known peptide substitutions affecting natural or in vitro selected mutants of EV-A71 and which act either on the adaptation of viruses in cell culture or on their virulence for animal models that could simulate virulence in humans. This article takes into account the published works exhaustively and gives a clear, precise and synthetic report. This work is of obvious importance for any virologist who would like to have an idea on the advancement of knowledge of the determinants which can make a viral strain of EV-A71 a virulent strain either for the cell and / or for the infected organism.

Among other things this study highlights in a particularly convincing way, the need to differentiate the viral determinants which promote the interaction of the virus with the residues of Heparan Sulfate on the surface of the cells which promote penetration and replication of the virus in certain cultured cells from those which affect the virulence of the virus in vivo in the infected host.

This review will surely be appreciated by enterovirologists but also by specialists of other viruses, with regard to studies aimed at determining why one viral strain is more virulent or more pathogenic than another. In addition, this review offers innovative methodological principles to develop future studies on the molecular determinants of the pathogenicity of EV-A71 or other viruses.

Author Response

Reviewer 1

The review by K. Kobayashi and S Koike aims to assess the known peptide substitutions affecting natural or in vitro selected mutants of EV-A71 and which act either on the adaptation of viruses in cell culture or on their virulence for animal models that could simulate virulence in humans. This article takes into account the published works exhaustively and gives a clear, precise and synthetic report. This work is of obvious importance for any virologist who would like to have an idea on the advancement of knowledge of the determinants which can make a viral strain of EV-A71 a virulent strain either for the cell and / or for the infected organism.

Among other things this study highlights in a particularly convincing way, the need to differentiate the viral determinants which promote the interaction of the virus with the residues of Heparan Sulfate on the surface of the cells which promote penetration and replication of the virus in certain cultured cells from those which affect the virulence of the virus in vivo in the infected host.

This review will surely be appreciated by enterovirologists but also by specialists of other viruses, with regard to studies aimed at determining why one viral strain is more virulent or more pathogenic than another. In addition, this review offers innovative methodological principles to develop future studies on the molecular determinants of the pathogenicity of EV-A71 or other viruses.

Response: We are delighted that Reviewer 1 evaluates this article highly. Although it seems that previous reports on the virulence of EV-A71 are inconsistent, we realized that they are actually consistent when viewed from the perspective of adaptation, which led us to write this review. We hope that this review will serve as an opening for a new stage in EV-A71 virulence research and that it will have a positive impact on other virologists.

Reviewer 2 Report

In this review, K. Kobayashi and S. Koike present current knowledge on enterovirus-A71 (EV-A71) adaptation and virulence. In particular, they highlight the impact of in vitro adaptation to heparan sulfate (HS) as an attachment receptor on in vivo attenuation. Both authors have previously reported attenuation of  EV-A71 HS-binding variants in animal models in numerous articles or reviews. Here they also describe the adaptation of EV-A71 in mice as well as other attenuation mechanisms such as polymerase fidelity and thermal stability. Although these topics have been covered previously, the review provides an interesting and comprehensive summary of studies on the subject, with useful tables.

Other comments:

  • Enteroviruses are increasingly referred to as genotypes rather than serotypes as sequencing has replaced viral typing by serology. The reviewer therefore suggests to replace “serotype” by “genotype”
  • 25: “EV-A71 is one of these” is not really appropriate.
  • 67: adapted rather than “adaptive” variants
  • In the description of the different animal models (monkeys, suckling mice…), the authors should also indicate the route of inoculation used.
  • 112: mice are “superior” model animals: this is a bit misleading.
  • The authors mention that the hSCARB2-tg mice is the best model but they must point out that in this model, endogenous mSCARB2 is still expressed and may have an impact on the outcome of the disease if the strains have the ability to bind to both human and mice SCARB2.
  • Since the authors discuss the role of heparan sulfate, PSGL-1 and SCARB2 binding in virulence in humans, mice or monkeys, they should include a table describing the capsid residues (if known) involved in binding to each of these receptors for each of these species. Similarly, a brief description of the available data on tissue distribution and structure of heparan sulfate in humans or in mice should be added.
  • 184 to 187: the reviewer agrees with the authors about adaptation bias. However, if a study describes an association between VP1-145G/Q/R and increased virulence in humans and an association between VP1-145E and mild cases, it is difficult to explain that in vitro adaptation occurs only for isolates from severe cases. All isolates should become HS-dependent in culture, regardless of the observed severity in humans.
  • For clarity reason, the table 1 should be presented differently: the first column should indicate the host and reports from human data should be grouped. In addition, the lines should be more clearly separated.
  • 224-229: While most isolates sequenced in GenBank have VP-145E, this does not mean that isolates with VP1-145G/Q are less frequently associated with severe disease. Since most EV-A71 infections are indeed mild, it is expected that most of the sequenced isolates are from mild cases. Furthermore, since many laboratories sequence EV-A71 after isolation in culture, it is rather surprising not to see many more isolates with VP1-145Q/G. To make their case, the authors should provide more details on the severity associated with the different variants sequenced and the method of viral amplification (with or without culture isolation).
  • Victorio et al (Emerging Microbes & Infections 2016) highlight that VP1 98E, 145A and 169F are involved in binding murine SCARB-2. This should be discussed in this review and the impact of these mutations should be included in figure 3
  • 368-370: did the authors also check for other adaptive mutations? Any mutation can affect in vivo virulence. The way inoculums are standardized for animal infections should also be discussed. The relative titer of different viral strains can vary completely from one cell line to another, which can distort in vivo results.

Author Response

Reviewer 2

In this review, K. Kobayashi and S. Koike present current knowledge on enterovirus-A71 (EV-A71) adaptation and virulence. In particular, they highlight the impact of in vitro adaptation to heparan sulfate (HS) as an attachment receptor on in vivo attenuation. Both authors have previously reported attenuation of EV-A71 HS-binding variants in animal models in numerous articles or reviews. Here they also describe the adaptation of EV-A71 in mice as well as other attenuation mechanisms such as polymerase fidelity and thermal stability. Although these topics have been covered previously, the review provides an interesting and comprehensive summary of studies on the subject, with useful tables.

Other comments:

  • Enteroviruses are increasingly referred to as genotypes rather than serotypes as sequencing has replaced viral typing by serology. The reviewer therefore suggests to replace “serotype” by “genotype”

Response 1: A recent article on the classification of Enteroviruses (1) uses the term "serotype," so "serotype" is used in this paper.

1: Simmonds P, Gorbalenya AE, Harvala H, Hovi T, Knowles NJ, Lindberg AM,

Oberste MS, Palmenberg AC, Reuter G, Skern T, Tapparel C, Wolthers KC, Woo PCY,

Zell R. Recommendations for the nomenclature of enteroviruses and rhinoviruses.

Arch Virol. 2020 Mar;165(3):793-797. DOI: 10.1007/s00705-019-04520-6. Erratum

in: Arch Virol. 2020 Jun;165(6):1515. PMID: 31980941; PMCID: PMC7024059.

  • 25: “EV-A71 is one of these” is not really appropriate.

Response 2: We modified Lines 24 – 25 of the revised manuscript according to the reviewer’s comment.

  • 67: adapted rather than “adaptive” variants

Response 3: We modified Line 70 of the revised manuscript according to the reviewer’s comment. Line 17 was also modified for the same reason.

  • In the description of the different animal models (monkeys, suckling mice…), the authors should also indicate the route of inoculation used.

Response 4: We have added descriptions of inoculation routes for various animal models. (Lines 108, 123, and 135 – 136 of the revised manuscript)

  • 112: mice are “superior” model animals: this is a bit misleading.

Response 5: We corrected "superior" to "useful." (Line 114 of the revised manuscript)

  • The authors mention that the hSCARB2-tg mice is the best model but they must point out that in this model, endogenous mSCARB2 is still expressed and may have an impact on the outcome of the disease if the strains have the ability to bind to both human and mice SCARB2.

Response 6: The concern by reviewer 2 is not necessary to be considered due to the following two reasons. First, it is not proved that suckling mice are infected via mouse Scarb2. Second, hSCARB2-tg mice are susceptible at any age, while non-tg suckling mice are only susceptible in a few weeks of birth, even if this infection is mediated by mouse Scarb2. When hSCARB2-tg mice older than three weeks old are used, muscle-tropic infections observed in neonatal mice are not observed anymore. Since our experiments usually use 6–7-week-old hSCARB2-tg mice, we think that the effect of mouse Scarb2 is negligible. For more details, please refer to our previous review article (2). We added descriptions of these. (Lines 139 – 144 and 387 of the revised manuscript)

  1. Kobayashi K, Koike S. 2020. Cellular receptors for enterovirus A71. J Biomed Sci 27:23.

  • Since the authors discuss the role of heparan sulfate, PSGL-1 and SCARB2 binding in virulence in humans, mice or monkeys, they should include a table describing the capsid residues (if known) involved in binding to each of these receptors for each of these species. Similarly, a brief description of the available data on tissue distribution and structure of heparan sulfate in humans or in mice should be added.

Response 7: We think that presenting a new table describing the binding site is not helpful for understanding. Because there is no report of the binding site on monkey SCARB2 and PSGL1, and mouse Scarb2 and Psgl1 cannot bind EV-A71. The binding sites on EV-A71 capsids for HS and PSGL-1 have already been written in the text (lines 151 – 152 and 174 – 175 of the revised manuscript), and the binding sites for SCARB2 have been added to the text (lines 49 – 51 of the revised manuscript). The differences in SCARB2 between human and mouse have already been described (lines 250 – 251 of the revised manuscript). We added the differences in PSGL1 between human and mouse (lines 62 – 63 of the revised manuscript). For HS, there are no differences among species, so we do not describe them. The differences in tissue distribution of each receptor have already been described in the text (lines 60 – 61 and 221 – 225 of the revised manuscript).

  • 184 to 187: the reviewer agrees with the authors about adaptation bias. However, if a study describes an association between VP1-145G/Q/R and increased virulence in humans and an association between VP1-145E and mild cases, it is difficult to explain that in vitro adaptation occurs only for isolates from severe cases. All isolates should become HS-dependent in culture, regardless of the observed severity in humans.

Response 8: We think that the time required for a mutant to appear in cultured cells and be selected varies to some extent. According to our unpublished results, when we isolated clinical isolates in normal cells, we found various strains with and without HS binding mutation. Therefore, if the number of passages is not too large, adaptation is not always detected clearly in the growth of cell culture systems. It is unclear why adaptation occurred more frequently only in samples of severe cases since it is unclear under what conditions (cell type and the number of passages) the virus was cultured in the studies that examined patient severity and viral genome sequence. In any case, we think that it is difficult to directly accept those results because of the adaptation risk of growing them in cultured cells. We also added the following sentence because we are concerned that readers may misunderstand that adaptation to cultured cells always occurs. “Adaptation to cultured cells is thought to occur quickly when HS-adapted mutant happens to appear as a founder, but if the founder does not appear within the culture period, adaptation does not occur. The emergence of adapted mutants is determined stochastically.” (lines 170 – 173 of the revised manuscript).

  • For clarity reason, the table 1 should be presented differently: the first column should indicate the host and reports from human data should be grouped. In addition, the lines should be more clearly separated.

Response 9: As the reviewer pointed out, Table 1 is confusing, so we modified it. The rows of amino acid mutations found in human patients are grouped at the top of the table and separated by a double line from the rest of the rows. I didn't add the column because the word "human" is written in the Description column.

  • 224-229: While most isolates sequenced in GenBank have VP-145E, this does not mean that isolates with VP1-145G/Q are less frequently associated with severe disease. Since most EV-A71 infections are indeed mild, it is expected that most of the sequenced isolates are from mild cases. Furthermore, since many laboratories sequence EV-A71 after isolation in culture, it is rather surprising not to see many more isolates with VP1-145Q/G. To make their case, the authors should provide more details on the severity associated with the different variants sequenced and the method of viral amplification (with or without culture isolation).

Response 10: As the reviewer points out, we think that there is a leap of logic in saying that VP1-145G/Q is an artifact and has nothing to do with outbreaks because VP1-145E is the majority in databases and epidemiological studies. This paragraph has been deleted.

  • Victorio et al (Emerging Microbes & Infections 2016) highlight that VP1 98E, 145A and 169F are involved in binding murine SCARB-2. This should be discussed in this review and the impact of these mutations should be included in figure 3

Response 11: Victorio et al. reported that VP1-K98E, E145A, and L169F allow the use of mouse SCARB2. Interestingly, a single amino acid mutation in VP1-L169F gives it infectivity in a mouse cell line. Although these amino acids are not direct binding sites for SCARB2, these changes may indirectly affect the binding to mouse SCARB2. However, little analysis of this amino acid has been done by other groups, and further analysis is required. Miyamura et al. also obtained a mutant that was adapted to L-PSGL1 cells without the VP1-K149I mutation. This mutant contained VP2-K69R, VP2-V135I, VP2-T176P, VP1-X145Q (where X is a mixture of two or more amino acids), and VP1-I249V. The rodent adaptation may occur in several different mechanisms. These reports are not included in Figure 3 because there are many uncertainties, but they are mentioned in the text. (Line 272-281 of the revised manuscript)

  • 368-370: did the authors also check for other adaptive mutations? Any mutation can affect in vivo virulence. The way inoculums are standardized for animal infections should also be discussed. The relative titer of different viral strains can vary completely from one cell line to another, which can distort in vivo results.

Response 12: We checked all known HS-binding and mouse-adapted mutations by NGS. This point has been added to the text. (Line 385 – 386 of the revised manuscript)

We also propose defining the viral dose for inoculation by RNA copy number or the number of infectious particles. In this case, one can avoid the problem that the relative titer of inoculation virus varies depending on the cell lines used for titration. (Line 392 – 395 of the revised manuscript)